# Towards Benchmarking Time Series Foundation Models on Native Scientific Data

**Lewis O'Donnell** [1]  **Arinbjörn Kolbeinsson** [2]  **Benedikt Kolbeinsson** [2]  **Marc Peter Deisenroth** [1]

## Abstract

Time series foundation models are evaluated on benchmarks that pre-align observations to a uniform grid, discarding the irregular, multi-rate structure present in real scientific data. We show empirically that preprocessing-induced degradation is large, architecture-specific, and invisible under conventional benchmarks. We propose a cross-domain benchmark spanning nuclear fusion, healthcare, and climate, stored in a unified observation set format that preserves native timestamps and mixed modalities without grid alignment and define four evaluation tasks and a hierarchical metric system to assess model capabilities.

## 1. Introduction

Foundation models for time series are evolving rapidly. Chronos (Ansari et al., 2024) (Ansari et al., 2025), MOIRAI (Woo et al., 2024) (Liu et al., 2024), MOMENT (Goswami et al., 2024), TimesFM (Das et al., 2024), and TiRex (Auer et al., 2025b;a) all demonstrate strong zero-shot and few-shot forecasting performance across diverse real-world datasets. These advances have been coupled with the development of a growing benchmarking infrastructure: GIFT-Eval (Aksu et al., 2024) curates 23 datasets spanning 7 domains, TFB (Qiu et al., 2024) profiles forecasting methods across 10 domains, and Time-Series-Library (Wang et al., 2024; Wu et al., 2023) provides evaluation across 5 tasks.

These benchmarks are well-designed for the problems they address. However, they all share a structural commitment that is rarely made explicit; all observations are aligned to a dense, regular, shared time grid before any model sees the data. This is appropriate for certain domains where measurements are generated on a fixed schedule. But in scientific domains such as healthcare, climate, and nuclear fusion, measurements are generated by heterogeneous instruments at different, often irregular rates, sensors drop

out mid-experiment, and some modalities produce spatially resolved fields or images rather than scalar values. Forcing this data onto a common grid is not neutral; it discards temporal structure, introduces false dependencies through interpolation, and causes information loss before any model is trained or evaluated, which can fundamentally alter downstream model rankings in ways standard evaluation cannot detect (Kazijevs & Samad, 2023b; Frost et al., 2026).

Benchmarks that pre-align data to a uniform grid incentivise architectures that expect that format, such as patch-based tokenisation and discrete value quantisation (Kottapalli et al., 2025), which in turn, define what benchmark performance means. Models that could process data at its native resolution, such as Neural ODE-based approaches like MIRA (Li et al., 2025) that directly ingest irregular time gaps and varying frequencies, remain under–investigated at scale (Liang et al., 2024) precisely because no multi-domain benchmark evaluates them in that setting.

To address this gap, we propose a benchmark comprising three open-source scientific datasets: MAST Tokamak plasma shots (Jackson et al., 2025; 2024), MIMIC-IV ICU stays (Johnson et al., 2024), and TC PRIMED tropical cyclones (Razin et al., 2023), stored in a unified observation set format that preserves native timestamps, per-signal missingness, and mixed modalities without any grid alignment. We define four evaluation tasks (forecasting, offline imputation, causal imputation, and cross-modal prediction) and a hierarchical, scale-normalised metric system that prevents high-frequency signals or data-rich domains from dominating aggregate scores. To motivate the need for native-resolution evaluation, we run a grid-sweep experiment holding the query set fixed while progressively coarsening the context, demonstrating that preprocessing-induced performance degradation is large, architecture-specific, and would be invisible under any benchmark that treats grid alignment as a fixed design constant rather than a modelling variable. Our contributions are: **(I)** A unified observation set data representation for heterogeneous, multi-rate, irregularly sampled scientific measurements (Section 2.1). **(II)** A cross-domain benchmark suite with four formally specified tasks and a five-level hierarchical metric (Section 2.2–2.4). **(III)** An empirical demonstration that uniform-grid preprocessing causes significant, model-dependent performance degradation (Section 3).

[1]UCL Centre for Artificial Intelligence [2]K01, Iceland. Correspondence to: Lewis O'Donnell <lewis.o'donnell.25@ucl.ac.uk>.

*Proceedings of the 2nd ICML Workshop on Foundation Models for Structured Data*, Seoul, South Korea. 2026. Copyright 2026 by the author(s).

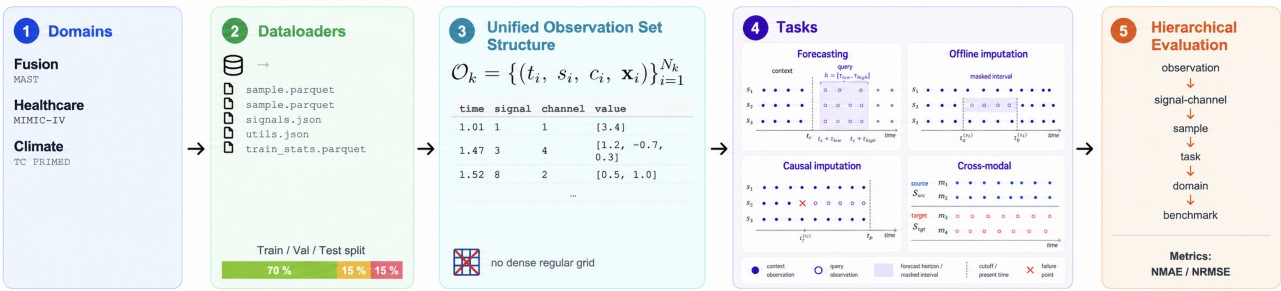

*Figure 1.* The benchmark pipeline. Raw scientific data from three domains is stored in a unified observation set format. Standardised dataloaders expose four evaluation tasks, each scored via a five-level hierarchical metric.

## 2. Methods

### 2.1. Unified Data Structure

A central challenge in benchmarking scientific time series is that the raw measurements are not naturally organised on a shared dense time grid; irregular sampling, per-signal missingness, mixed dimensionality, and static metadata are all common. We therefore use an observation set representation that stores the data at its native resolution.

Building on the work (Horn et al., 2020), each experimental sample, (a plasma shot, ICU stay, storm), $k$, is a finite collection of observations:

$$\mathcal{O}_k = \left\{ (t_i,\ s_i,\ c_i,\ \mathbf{x}_i) \right\}_{i=1}^{N_k}$$

where $t_i \in \mathbb{R}$ is the continuous timestamp of the $i$-th observation relative to the start of the sample, $s_i \in \{1, \ldots, S\}$ identifies the signal type (e.g. "flux loop", "heart rate"), $c_i \in \{1, \ldots, C_{s_i}\}$ identifies the channel within that signal type (e.g. which magnetic probe), and $\mathbf{x}_i \in \mathbb{R}^{d_{s_i}}$ is the observed value. The signal type $s_i$ determines the meaning, units, and value dimension of the observation. For scalar signals, $d_{s_i} = 1$, whereas profile or field measurements are stored as vector-valued observations, with $d_{s_i}$ equal to the number of flattened spatial degrees of freedom.

By choosing this representation, we do not impose a global sampling grid or shared observation schedule. For a fixed signal-channel pair $(s, c)$, the corresponding observed sequence is:

$$\mathcal{O}_{k,s,c} = \left\{ (t_i, \mathbf{x}_i) : (t_i,\ s_i,\ c_i,\ \mathbf{x}_i) \in \mathcal{O}_k, s_i = s, c_i = c \right\}$$

Mixed-frequency sampling appears as different cardinalities and inter-observation intervals across signal-channel pairs. Irregular sampling appears as non-constant gaps between consecutive timestamps within $\mathcal{O}_{k,s,c}$. Missingness is represented by the absence of observations, rather than the insertion of imputed values or padding. Thus, the number of observations, $N_k$, varies across samples, reflecting differences in duration, instrumentation coverage, and sampling frequency.

Static data is represented as pseudo-observations with a distinguished timestamp, $t_{\text{static}}$, and signal type, $s_{\text{static}}$:[1]

$$\mathcal{S}_k = \left\{ (t_{\text{static}},\ s_{\text{static}}, c_i, \mathbf{x}_i) \right\}_{i=1}^{N_k^{\text{static}}}$$

Where $N_k^{\text{static}}$ is the total number of static channels for sample $k$. The full sample is then: $\tilde{\mathcal{O}}_k = \mathcal{O}_k \cup \mathcal{S}_k$

This format preserves native measurements and leaves any resampling, interpolation, padding or gridding as an explicit modelling choice rather than a hidden preprocessing assumption.

### 2.2. Three Scientific Domains

The benchmark comprises three open-source scientific datasets that naturally exhibit the observation set structure. Namely we use the MAST dataset (*nuclear fusion*) (Jackson et al., 2025; 2024), the MIMIC-IV dataset (*healthcare*) (Johnson et al., 2024), and the TC PRIMED dataset (*climate*) (Razin et al., 2023). In the fusion domain, each sample is a MAST plasma shot, where observations correspond to diagnostic measurements such as magnetic probes, plasma current, density profiles, and equilibrium quantities. In healthcare data, each sample is an ICU stay from MIMIC, where observations include irregularly sampled vital signs, laboratory measurements, interventions, and static patient metadata. For climate, each sample is a tropical cyclone from TC PRIMED, with observations including storm-track variables, environmental diagnostics, and satellite-derived overpass measurements that are aperiodic and sensor-dependent. We split samples in each dataset into training, validation and test sets $(70\%, 15\%, 15\%)$(see Appendix E)

### 2.3. Task Definitions

We define four benchmark tasks that assess the capabilities of scientific time-series foundation models, namely:

---

[1]We note here that in our formulation $t_{\text{static}} = -1$ and $s_{\text{static}} = 0$ for all static data. Researchers implementing models should treat $s_i = 0$ as a distinct token type rather than a temporal observation. Static data may include, for example, patient demographics, machine configuration, or storm metadata.

forecasting, offline imputation, causal imputation and cross-modal prediction. Each task is specified by splitting test samples into two disjoint sets (that need not exhaust the full sample): a context set, $\mathcal{C}_k$, which is observed by the model (static observations are included in the context set for each task), and a query set, $\mathcal{Q}_k$, whose values, $\mathbf{x}_i$, must be predicted.

For each query observation, $(t_i, s_i, c_i, \mathbf{x}_i) \in \mathcal{Q}_k$, the model receives the context, $\mathcal{C}_k$, and the query coordinates $(t_i, s_i, c_i)$ and predicts $\hat{\mathbf{x}}_i = f_\theta(\mathcal{C}_k, t_i, s_i, c_i)$. We now describe the formulation of each task.

**Forecasting.** Forecasting evaluates a model's ability to predict future observations from past observations. Given a cut-off time $t_c$ and forecast horizon $h = [\tau_{low}, \tau_{high}]$, with $0 \leq \tau_{low} \leq \tau_{high}$. The context contains all temporal observations before $t_c$, together with the static observations, and the query set contains all observations whose timestamps fall inside the forecast horizon.

$$\mathcal{C}_k = \mathcal{S}_k \cup \{(t_i, s_i, c_i, \mathbf{x}_i) \in \mathcal{O}_k : t_i < t_c\}$$
$$\mathcal{Q}_k = \{(t_i, s_i, c_i, \mathbf{x}_i) \in \mathcal{O}_k : t_i - t_c \in h\}$$

The horizon band need not start at $t_c$, allowing evaluation of delayed or multi-step horizons. Observations outside the horizon band are not used for scoring.

**Offline Imputation** Offline imputation tests whether a model can reconstruct missing portions of partially observed samples by using information from both before and after the missing interval. We define a set of held-out signals $\mathcal{H} \subset \{1, \ldots, S\}$ and for each $s \in \mathcal{H}$, there is a masked time interval $[t_a^{(s)}, t_b^{(s)}]$. The query set is the set of observations from held-out signals that lie inside their masked interval, and the context set contains all remaining temporal observations and static data:

$$\mathcal{Q}_k = \{(t_i, s_i, c_i, \mathbf{x}_i) \in \mathcal{O}_k : s_i \in \mathcal{H} \wedge t_i \in [t_a^{(s_i)}, t_b^{(s_i)}]\}$$
$$\mathcal{C}_k = \mathcal{S}_k \cup (\mathcal{O}_k \backslash \mathcal{Q}_k)$$

**Causal Imputation.** While offline imputation allows a model to use observations before and after the masked interval, causal imputation evaluates the same reconstruction ability under a real-time constraint. That is, the model is restricted to information available up to the present time $t_p$. Let $\mathcal{H}$ be the set of held out signals, and let $t_f^{(s)}$ denote the failure time for each $s \in \mathcal{H}$, with $t_f^{(s)} < t_p$. The context set contains all observations up to the present time, the query set contains the unavailable observations from the failed signals between failure and the present:

$$\mathcal{C}_k = \mathcal{S}_k \cup \{(t_i, s_i, c_i, \mathbf{x}_i) \in \mathcal{O}_k : t_i \leq t_p \wedge (s_i \notin \mathcal{H} \vee t_i < t_f^{(s_i)})\}$$
$$\mathcal{Q}_k = \{(t_i, s_i, c_i, \mathbf{x}_i) \in \mathcal{O}_k : s_i \in \mathcal{H} \wedge t_i \in [t_f^{(s_i)}, t_p]\}$$

Thus, observations after $t_p$ are excluded from both context and query, enforcing the causal setting.

**Cross Modal Prediction.** Cross modal prediction examines whether a model can infer target measurements from different source measurements. Let $\mathcal{S}_{src} \subset \{1, \ldots, S\}$ and $\mathcal{S}_{tgt} \subset \{1, \ldots, S\}$ be disjoint sets of source and target signals. The context contains all observations from the source signals and the query contains all observations from target signals:

$$\mathcal{C}_k = \mathcal{S}_k \cup \{(t_i, s_i, c_i, \mathbf{x}_i) \in \mathcal{O}_k : s_i \in \mathcal{S}_{src}\}$$
$$\mathcal{Q}_k = \{(t_i, s_i, c_i, \mathbf{x}_i) \in \mathcal{O}_k : s_i \in \mathcal{S}_{tgt}\}$$

This task tests whether a model is able to learn relationships between different measurement modalities, rather than only temporal persistence within individual signals.

## 2.4. Metrics

The benchmark spans domains with different physical units, sampling frequencies, signal dimensions, and sample lengths. Hence, a raw error averaged over all observations would be dominated by high-frequency signals and long samples. Therefore, we use scale-normalised errors computed with statistics from the training set only, followed by a hierarchical aggregation.

Let $(t_i, s_i, c_i, \mathbf{x}_i) \in \mathcal{Q}_k$ be an observation from the query set, let $\hat{\mathbf{x}}_i$ be a model's prediction, and let $\sigma_{s_i, c_i, m}$ be the training set standard deviation of component $m$ for signal $s_i$, channel $c_i$. We compute the observation level normalised absolute and squared errors by averaging over all of the $m$ components of $\mathbf{x}_i$:[2]

$$\text{NMAE}_i = \frac{1}{d_{s_i}} \sum_{m=1}^{d_{s_i}} \frac{|x_{i,m} - \hat{x}_{i,m}|}{\sigma_{s_i, c_i, m}}, \quad \text{NMSE}_i = \frac{1}{d_{s_i}} \sum_{m=1}^{d_{s_i}} \left(\frac{x_{i,m} - \hat{x}_{i,m}}{\sigma_{s_i, c_i, m}}\right)^2$$

To prevent dense signals from dominating the benchmark, we aggregate errors hierarchically by first averaging within each signal-channel pair, $(s, c)$, in a sample. This gives per-signal-channel NMAE and NRMSE for a sample, where NRMSE is the square root of the mean NMSE for that signal-channel pair. We then proceed by averaging these group-level scores equally across signal channel pairs within a sample, then samples within a task, then tasks within a domain and finally domains within the benchmark (see Appendix A for full hierarchical metric definitions). This ensures that each level of the hierarchy contributes equally to the final benchmark score. The overall benchmark reports two primary metrics, NMAE and NRMSE, while also retaining the full hierarchy for diagnosis by domain, task, signal, and channel. Baseline model scores are reported in Appendix C.

---

[2]Note here that degenerate standard deviations are replaced by a small positive constant to avoid dividing by zero.

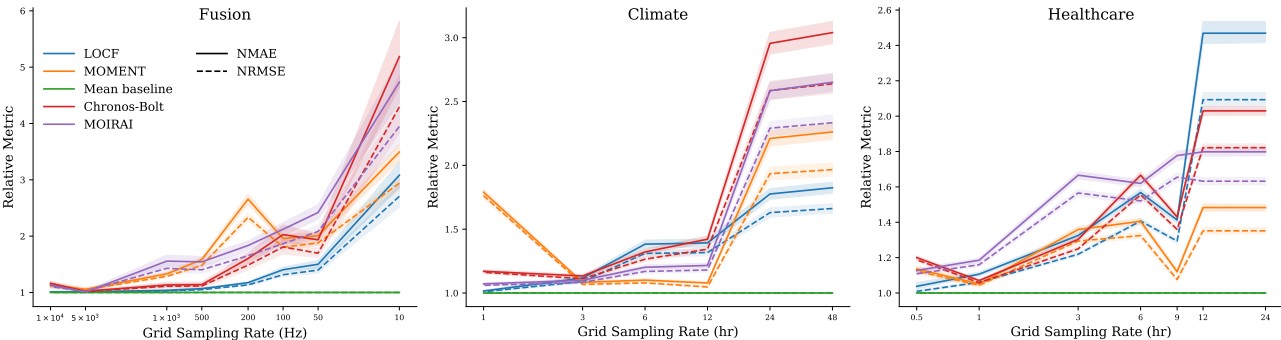

*Figure 2.* Representation commitment cost across three domains. $R(g) = 1$ indicates no degradation relative to native resolution; values above 1 quantify performance lost to grid preprocessing. The mean baseline satisfies $R_{\text{mean}}(g) = 1$ exactly under all conditions.

## 3. Results

We perform a diagnostic grid-sweep experiment to measure the representation commitment cost: the performance distortion introduced when native scientific time series are forced onto a uniform grid before modelling. For each test sample, the forecasting query set and evaluation metric are held fixed while only the context representation is varied, from native irregular timestamps to increasingly coarse uniform grids. We define the relative metric

$$R(g) = \frac{1}{|\mathcal{K}|} \sum_{k \in \mathcal{K}} \frac{1}{|S_k|} \sum_{s \in S_k} \frac{\mathcal{L}_{k,s}^{\text{NMAE}}(g)}{\mathcal{L}_{k,s}^{\text{NMAE}}(\text{native})}$$

where $\mathcal{L}_{k,s}^{\text{NMAE}}(g)$ is the signal-level NMAE (see Section 2.4 and Appendix A) under grid condition $g$ and $\mathcal{K}$ is the test set. $R(g) = 1$ indicates no degradation, and values above 1 quantify information lost due to preprocessing. We evaluate two baselines (mean and LOCF (Last Observation Carried Forward)) and three pre-trained foundation models (MOMENT, Chronos-Bolt, MOIRAI). The mean baseline is context-independent, so $R(g) = 1$ under all conditions, confirming that any observed change is attributable solely to the context representation. The results are shown in Figure 2 and in detail in Appendix B.

Figure 2 shows that degradation is large, architecture-specific, and qualitatively distinct across domains. In healthcare, all context-dependent models degrade as the grid coarsens. By 12h all foundation models have $R > 1.60$ and have collapsed to the mean baseline, indicating that a 12-hour grid discards sufficient temporal structure that no context-dependent model can outperform predicting the training mean. In climate, degradation is strong and monotonic; Chronos-Bolt and MOIRAI reach $R = 3.04$ and $R = 2.65$, respectively, at the 48h grid as the context loses the sub-daily structure on which storm-track forecasting relies. In fusion, LOCF degrades gradually on fast signals, reaching $R = 3.04$ at 10Hz. With Chronos fast signals degrading to $R = 5.61$ at 10Hz, a substantially larger ef-

fect than observed in healthcare or climate, reflecting the extreme multi-rate structure of fusion diagnostics spanning five orders of magnitude in sampling frequency. Collectively, these results demonstrate that grid choice is not neutral: its effects differ in magnitude and kind across models and domains, and expose failure modes that conventional benchmarks cannot detect.

## 4. Related Work

Existing time series foundation models (Ansari et al., 2024; Woo et al., 2024; Goswami et al., 2024) and benchmarks (Aksu et al., 2024; Qiu et al., 2024) share a structural commitment to pre-aligned, uniform frequency grids, with preprocessing choices that are invisible to evaluation. Models capable of native resolution processing exist and show promise (Li et al., 2025; Boschi et al., 2026), but the benchmarking infrastructure to evaluate them at scale and across domains is nascent. The closest benchmark is Tokamark (Rousseau et al., 2026), which is fusion-specific and evaluates two models. Our benchmark addresses this directly by treating resampling as a modelling variable rather than a design constant, across three heterogeneous scientific domains. A comprehensive review of general and domain-specific benchmarks is given in Appendix D.

## 5. Conclusions and Future Work

We have presented a cross-domain benchmark for scientific time series that evaluates models at native resolution across nuclear fusion, healthcare, and climate, without imposing grid alignment as a preprocessing constant. We demonstrate that this alignment is not neutral. Preprocessing-induced degradation is large, architecture-specific, and invisible under conventional benchmarks. Future work includes evaluating models that natively handle irregular data, such as Neural ODE-based approaches, expanding to additional scientific domains, introducing more tasks, and finally open-sourcing our code.

## Acknowledgements

This work was supported by the UK Engineering and Physical Sciences Research Council (EPSRC) grant EP/Y034767/1 and Iceland Technology Development Fund grant 2525674-601. We also acknowledge Samuel Jackson (*United Kingdom Atomic Energy Authority*) and James Hetherington (*UCL Advanced Research Computing Centre*) for their thoughtful discussions in the early stages of the work.

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

# A. Full Metric Definition

This appendix gives the full formal definition of the benchmark metrics described in Section 2.4.

Let $\mathcal{Q}_k$ be the query set for sample $k$. For each signal-channel pair $(s,c)$, define the subset of query observations $\mathcal{Q}_{k,(s,c)} = \{(t_i, s_i, c_i, \mathbf{x}_i) \in \mathcal{Q}_k : s_i = s, \; c_i = c\}$, and let $I_k = \{(s,c) : |\mathcal{Q}_{k,(s,c)}| > 0\}$ be the set of signal-channel pairs that appear in the query set for sample $k$.

For every signal $s$, channel $c$, and component $m$, we compute the training set mean and standard deviation, $\mu_{s,c,m}$ and $\sigma_{s,c,m}$, using only the training split. These quantities are fixed before evaluation and are used for all validation and test samples. If $\sigma_{s,c,m}$ is zero or non-finite, it is replaced by a small positive constant $\epsilon$.

If a model does not return a prediction for a query observation, or returns a prediction with the wrong dimensionality, we evaluate that observation using the training-set mean prediction for the corresponding signal-channel pair $\hat{\mathbf{x}}_{i,m} = \mu_{s,c,m}$. This prevents models from improving their score by omitting difficult query points.

## A.1. Observation-Level Errors

For a query observation $(t_i, s_i, c_i, \mathbf{x}_i) \in \mathcal{Q}_k$, with prediction $\hat{\mathbf{x}}_i$, define the observation-level normalised mean absolute error as

$$\ell_i^{\text{NMAE}} = \frac{1}{d_{s_i}} \sum_{m=1}^{d_{s_i}} \frac{|x_{i,m} - \hat{x}_{i,m}|}{\sigma_{s_i,c_i,m}},$$

and the observation-level normalised mean squared error as

$$\ell_i^{\text{NMSE}} = \frac{1}{d_{s_i}} \sum_{m=1}^{d_{s_i}} \left( \frac{x_{i,m} - \hat{x}_{i,m}}{\sigma_{s_i,c_i,m}} \right)^2.$$

## A.2. Signal-Channel-Level Errors

For each signal-channel pair $(s,c)$ in sample $k$, we average over the query observations belonging to that pair:

$$\mathcal{L}_{k,(s,c)}^{\text{NMAE}} = \frac{1}{|\mathcal{Q}_{k,(s,c)}|} \sum_{i \in \mathcal{Q}_{k,(s,c)}} \ell_i^{\text{NMAE}},$$

and

$$\mathcal{L}_{k,(s,c)}^{\text{NRMSE}} = \sqrt{\frac{1}{|\mathcal{Q}_{k,(s,c)}|} \sum_{i \in \mathcal{Q}_{k,(s,c)}} \ell_i^{\text{NMSE}}}$$

This is the first level of aggregation. Each signal-channel pair receives its own score before any averaging across the sample.

## A.3. Sample-Level Errors

The sample-level score is the average across signal-channel pairs that appear in the query set:

$$\mathcal{L}_k^{\text{NMAE}} = \frac{1}{|I_k|} \sum_{(s,c) \in I_k} \mathcal{L}_{k,(s,c)}^{\text{NMAE}},$$

and

$$\mathcal{L}_k^{\text{NRMSE}} = \frac{1}{|I_k|} \sum_{(s,c) \in I_k} \mathcal{L}_{k,(s,c)}^{\text{NRMSE}}.$$

Thus, each signal-channel pair contributes equally within a sample, regardless of how many raw observations it contains.

### A.4. Task-Level Errors

For a task $\tau$, let $T_\tau$ denote the set of evaluation samples for that task. The task-level metrics are

$$\mathcal{L}_\tau^{\text{NMAE}} = \frac{1}{|T_\tau|} \sum_{k \in T_\tau} \mathcal{L}_k^{\text{NMAE}},$$

and

$$\mathcal{L}_\tau^{\text{NRMSE}} = \frac{1}{|T_\tau|} \sum_{k \in T_\tau} \mathcal{L}_k^{\text{NRMSE}}.$$

### A.5. Domain-Level Errors

For a domain $d$, let $B_d$ denote the set of benchmark tasks evaluated for that domain. The domain-level metrics are

$$\mathcal{L}_d^{\text{NMAE}} = \frac{1}{|B_d|} \sum_{\tau \in B_d} \mathcal{L}_\tau^{\text{NMAE}},$$

and

$$\mathcal{L}_d^{\text{NRMSE}} = \frac{1}{|B_d|} \sum_{\tau \in B_d} \mathcal{L}_\tau^{\text{NRMSE}}.$$

### A.6. Benchmark-Level Errors

Finally, let $D$ denote the set of domains in the benchmark. The overall benchmark scores are

$$\mathcal{L}^{\text{NMAE}} = \frac{1}{|D|} \sum_{d \in D} \mathcal{L}_d^{\text{NMAE}},$$

and

$$\mathcal{L}^{\text{NRMSE}} = \frac{1}{|D|} \sum_{d \in D} \mathcal{L}_d^{\text{NRMSE}}.$$

This hierarchy ensures that the final score is not dominated by high-frequency signals, long samples, tasks with more query observations, or domains with more data.

# B. Granular Grid-Sweep Results

This appendix reports NMAE for each benchmark task across all combinations of domain, model, signal speed, and grid condition. Each table corresponds to one task and stacks the three domains vertically. The value columns are labelled by the native context together with progressively coarser uniform grids $g_1, \ldots, g_8$. The number of populated columns varies by domain because the physical grid rates differ. The mapping from $g_i$ to a physical frequency is given in each table caption.

**Grid conditions.**   Resampling levels were chosen to cover the sampling rates present in each domain. In fusion, equilibrium signals are sampled at 200 Hz, magnetics at 4–5 kHz, 50 kHz for soft X-ray and saddle probes, and 500 kHz for fast magnetic probes. The eight uniform-grid conditions $g_1$−$g_8$ (10 kHz down to 10 Hz) are chosen to progressively discard information across the sampling frequency range. For healthcare, native rates are approximately hourly (vital signs) and four-hourly (GCS, temperature). Seven levels span 30 min ($g_1$) to 24 h ($g_7$). For climate, the track and environmental diagnostic signals are sampled at 6 h; six levels span 1 h ($g_1$) to 48 h ($g_6$), where $g_6$ yields roughly one context observation per two days.

**Models and configurations.**   Five models are evaluated. *Mean baseline:* predicts the training-set mean for each signal-channel pair. *LOCF:* carries the last observed context value forward to each query timestamp *Chronos-Bolt-Small:* we use the `amazon/chronos-bolt-small` checkpoint (Ansari et al., 2025), producing a deterministic point estimate from its median-quantile output head. *MOIRAI-1.0-R-small:* Salesforce MOIRAI (size small, version 1.0) (Woo et al., 2024), drawing 100 Monte Carlo samples per query (sample mean used as the point estimate) with a context window of 200 grid steps. *MOMENT:* the `AutonLab/MOMENT-1-large` checkpoint (Goswami et al., 2024), with a randomly-initialised linear forecasting head fine-tuned on the domain training set using the Adam optimiser at learning rate $10^{-4}$ for 3 epochs (random seed 42 controls training-file selection and head initialisation; the same seed is used across all three domains). Training files are capped to control wall-clock time: 100 files for fusion (from 2 022 available), 5 000 for healthcare (from 52 380 available), and uncapped for climate.

Foundation models (Chronos, MOIRAI, MOMENT) are evaluated on the *forecasting* task only. The *offline imputation*, *causal imputation*, and *cross-modal* tasks use only the Mean and LOCF baselines, as these foundation models are causal forecasters that cannot condition on future observations. For the cross-modal task, LOCF has no access to a predecessor of any target signal in the context, so it falls back to the training-mean penalty and matches Mean exactly.

Each foundation model operates on a uniform grid internally, whereas the query set retains its native, irregular timestamps. For every query observation, we evaluate the model on the relevant grid and map its grid-indexed output back to the query time by nearest-grid-step assignment (Chronos, MOIRAI) or by truncating the fixed-horizon output to the query window (MOMENT, as above); a query time not covered by any grid step receives the training-mean. The query set and metric are thus identical across all grid conditions $g$, and only the context the model conditions on changes, which is what makes $R(g)$ well defined.

**Forecasting task configuration.**   Each domain uses a single forecasting task with context cut-off $t_c$ and query horizon band $[0, \tau_{\text{high}}]$ relative to $t_c$. For fusion, $t_c = 0.15$ s, $\tau_{\text{high}} = 0.05$ s (query window 150–200 ms from shot start). In healthcare, $t_c$ is the midpoint of each stay, $\tau_{\text{high}} = 12$ h. For climate, $t_c = 24$ h from storm genesis, $\tau_{\text{high}} = 6$ h. MOMENT requires a fixed discrete horizon $H$, set as $H = \lceil \tau_{\text{high}} \cdot f_{\text{finest}} \rceil$ where $f_{\text{finest}}$ is the frequency of the finest uniform-grid condition: $H = 500$ for fusion ($f_{\text{finest}} = 10$ kHz), $H = 24$ for healthcare ($f_{\text{finest}} = 1/(30 \text{ min})$), and $H = 6$ for climate ($f_{\text{finest}} = 1/\text{h}$). At coarser grid conditions, fewer than $H$ steps may fall in the query window; MOMENT outputs are truncated to the window and scored accordingly.

**Signal speed partition.**   Within each domain block, signals are partitioned by their empirical median inter-observation interval. Signals at or above the domain frequency threshold are classed as *fast*; those below are *slow*. Thresholds: $1/7{,}200$ Hz for healthcare; 1 kHz for fusion; $1/(3 \text{ h})$ for climate. TC-PRIMED results appear only under the slow partition: the 6-hourly track and environmental diagnostic signals constitute the primary channels for which the forecasting query set is populated; the irregularly-timed satellite overpass channels, while faster than the 3 h threshold, do not contribute query observations under the defined 6-h forecast horizon window.

**Uncertainty quantification.**   All cells for all models report a test-unit bootstrap 95% confidence interval (Mercado et al., 2025). We resample test units (shots, stays, or storms) with replacement, recompute the cell mean NMAE, and repeat $B = 1{,}000$ times. The reported interval is $\mu \pm h$, where $\mu$ is the point estimate and $h = \max(\mu - q_{2.5}, q_{97.5} - \mu)$ is the

larger percentile half-width. This procedure resamples the test units rather than the model and is therefore applicable to deterministic algorithms (Mean, LOCF) and pre-trained models (Chronos, MOIRAI, MOMENT) alike. For MOMENT, which undergoes a single fine-tuning run per domain, fine-tune variance is not separately quantified; the reported CI captures test-unit sampling variability only. A dash (–) denotes a condition not evaluated for that domain–task combination; in relative-metric tables the native column ($R = 1.00$ by construction) and the Mean baseline row ($R_{\mathrm{mean}} = 1.00$ by construction) are omitted.

The bold results are the best-performing models for each grid resolution, for both fast and slow signals, where necessary.

*Table 1.* Forecasting NMAE per (domain, model, signal speed) across grid coarsening levels $g_1$–$g_8$. $g_1$–$g_8$ map to: Healthcare = 30 min, 1 h, 3 h, 6 h, 9 h, 12 h, 24 h ($g_8$=–); Fusion = 10 kHz, 5 kHz, 1 kHz, 500 Hz, 200 Hz, 100 Hz, 50 Hz, 10 Hz; Climate = 1 h, 3 h, 6 h, 12 h, 24 h, 48 h ($g_7$–$g_8$=–). Cells show $\mu \pm h$ where $h$ is the 95 % percentile-bootstrap half-width over test stays ($B = 1{,}000$).

| Domain | Model | Speed | native | $g_1$ | $g_2$ | $g_3$ | $g_4$ | $g_5$ | $g_6$ | $g_7$ | $g_8$ |
|---|---|---|---|---|---|---|---|---|---|---|---|
| Healthcare (MIMIC-IV) | Mean | fast | 0.744±0.005 | 0.744±0.005 | 0.744±0.005 | 0.744±0.005 | 0.744±0.005 | 0.744±0.005 | **0.744±0.005** | **0.744±0.005** | – |
| | Mean | slow | 0.747±0.006 | 0.747±0.006 | 0.747±0.007 | 0.747±0.006 | 0.747±0.007 | 0.747±0.007 | 0.747±0.006 | 0.747±0.007 | – |
| | LOCF | fast | 0.556±0.005 | **0.534±0.005** | 0.543±0.005 | **0.577±0.005** | **0.629±0.005** | **0.581±0.005** | 0.855±0.008 | 0.855±0.008 | – |
| | LOCF | slow | **0.287±0.007** | **0.304±0.007** | **0.311±0.007** | **0.342±0.007** | **0.405±0.009** | **0.473±0.011** | **0.696±0.014** | **0.696±0.013** | – |
| | Chronos | fast | **0.511±0.004** | 0.535±0.004 | **0.513±0.004** | 0.586±0.005 | 0.693±0.006 | 0.617±0.005 | **0.744±0.005** | **0.744±0.005** | – |
| | Chronos | slow | 0.348±0.007 | 0.422±0.008 | 0.361±0.007 | 0.374±0.007 | 0.537±0.009 | 0.548±0.007 | 0.747±0.007 | 0.747±0.007 | – |
| | MOIRAI | fast | 0.585±0.005 | 0.568±0.005 | 0.593±0.005 | 0.761±0.006 | 0.749±0.006 | 0.776±0.006 | **0.744±0.005** | **0.744±0.005** | – |
| | MOIRAI | slow | 0.450±0.009 | 0.381±0.008 | 0.422±0.008 | 0.525±0.010 | 0.546±0.012 | 0.660±0.011 | 0.747±0.006 | 0.747±0.007 | – |
| | MOMENT | fast | 0.691±0.005 | 0.625±0.004 | 0.616±0.004 | 0.838±0.007 | 0.863±0.009 | 0.662±0.006 | **0.744±0.005** | **0.744±0.005** | – |
| | MOMENT | slow | 0.486±0.008 | 0.441±0.008 | 0.446±0.008 | 0.561±0.010 | 0.583±0.011 | 0.537±0.008 | 0.747±0.007 | 0.747±0.006 | – |
| Fusion (MAST) | Mean | fast | 0.455±0.015 | 0.455±0.015 | 0.455±0.015 | 0.455±0.014 | 0.455±0.015 | 0.455±0.016 | 0.455±0.015 | 0.455±0.015 | **0.455±0.015** |
| | Mean | slow | 0.683±0.078 | 0.683±0.084 | 0.683±0.079 | 0.683±0.082 | 0.683±0.079 | 0.683±0.078 | 0.683±0.077 | **0.683±0.080** | **0.683±0.077** |
| | LOCF | fast | 0.241±0.012 | 0.240±0.012 | 0.241±0.011 | 0.245±0.011 | 0.253±0.012 | 0.277±0.013 | 0.320±0.015 | 0.322±0.015 | 0.650±0.036 |
| | LOCF | slow | **0.481±0.070** | 0.507±0.084 | 0.508±0.085 | **0.516±0.080** | **0.524±0.081** | **0.567±0.084** | **0.638±0.072** | 0.752±0.085 | 0.944±0.088 |
| | Chronos | fast | **0.182±0.009** | **0.203±0.010** | **0.181±0.010** | **0.187±0.009** | **0.183±0.010** | **0.260±0.016** | **0.307±0.013** | **0.305±0.015** | 0.679±0.030 |
| | Chronos | slow | 0.590±0.082 | 0.515±0.079 | 0.558±0.081 | 0.528±0.082 | 0.611±0.073 | 0.653±0.072 | 0.700±0.080 | 0.742±0.078 | **0.683±0.074** |
| | MOIRAI | fast | 0.205±0.011 | 0.209±0.010 | 0.210±0.013 | 0.272±0.013 | 0.270±0.012 | 0.321±0.014 | 0.363±0.015 | 0.403±0.016 | 0.615±0.024 |
| | MOIRAI | slow | 0.680±0.065 | **0.491±0.085** | **0.501±0.075** | 0.594±0.074 | 0.677±0.067 | 0.727±0.072 | 0.772±0.067 | 0.762±0.065 | **0.683±0.081** |
| | MOMENT | fast | 0.399±0.015 | 0.406±0.012 | 0.412±0.018 | 0.495±0.019 | 0.495±0.019 | 0.813±0.034 | 0.637±0.036 | 0.645±0.023 | 0.729±0.038 |
| | MOMENT | slow | 1.153±0.096 | 0.893±0.069 | 0.766±0.082 | 0.755±0.080 | 0.835±0.072 | 1.161±0.092 | 1.191±0.119 | 1.007±0.123 | **0.683±0.081** |
| Climate (TC-PRIMED) | Mean | slow | 0.688±0.015 | 0.688±0.015 | 0.688±0.015 | 0.688±0.015 | 0.688±0.015 | 0.688±0.016 | 0.688±0.016 | – | – |
| | LOCF | slow | **0.430±0.010** | **0.424±0.009** | **0.434±0.009** | **0.488±0.011** | **0.489±0.011** | **0.560±0.014** | **0.569±0.014** | – | – |
| | Chronos | slow | 0.436±0.010 | 0.464±0.010 | 0.442±0.009 | 0.489±0.012 | 0.518±0.012 | 0.675±0.016 | 0.688±0.015 | – | – |
| | MOIRAI | slow | 0.447±0.009 | 0.448±0.009 | 0.466±0.011 | **0.488±0.012** | 0.495±0.012 | 0.676±0.015 | 0.688±0.015 | – | – |
| | MOMENT | slow | 0.513±0.011 | 0.735±0.012 | 0.530±0.012 | 0.539±0.014 | 0.530±0.013 | 0.676±0.015 | 0.688±0.016 | – | – |

*Table 2.* Forecasting NMAE per (domain, model) aggregated over fast and slow signal speeds. Coarsening mapping and CI notation as in Table 1.

| Domain | Model | native | $g_1$ | $g_2$ | $g_3$ | $g_4$ | $g_5$ | $g_6$ | $g_7$ | $g_8$ |
|---|---|---|---|---|---|---|---|---|---|---|
| Healthcare (MIMIC-IV) | Mean | 0.745±0.005 | 0.745±0.006 | 0.745±0.006 | 0.745±0.006 | 0.745±0.006 | 0.745±0.006 | **0.745±0.006** | **0.745±0.006** | – |
| | LOCF | **0.422±0.006** | **0.420±0.006** | **0.428±0.006** | **0.460±0.006** | **0.518±0.007** | **0.528±0.008** | 0.776±0.011 | 0.776±0.011 | – |
| | Chronos | 0.430±0.006 | 0.479±0.006 | 0.437±0.005 | 0.480±0.006 | 0.615±0.007 | 0.583±0.006 | **0.745±0.006** | **0.745±0.006** | – |
| | MOIRAI | 0.518±0.007 | 0.475±0.006 | 0.508±0.006 | 0.644±0.008 | 0.648±0.009 | 0.719±0.008 | **0.745±0.006** | **0.745±0.006** | – |
| | MOMENT | 0.589±0.006 | 0.533±0.006 | 0.531±0.006 | 0.700±0.006 | 0.723±0.009 | 0.600±0.006 | **0.745±0.006** | **0.745±0.006** | – |
| Fusion (MAST) | Mean | 0.569±0.046 | 0.569±0.049 | 0.569±0.047 | 0.569±0.048 | 0.569±0.047 | 0.569±0.046 | 0.569±0.045 | 0.569±0.047 | **0.569±0.045** |
| | LOCF | **0.361±0.040** | 0.374±0.047 | 0.374±0.047 | 0.380±0.045 | **0.388±0.046** | **0.422±0.048** | **0.479±0.043** | 0.537±0.050 | 0.797±0.061 |
| | Chronos | 0.386±0.045 | 0.359±0.045 | 0.370±0.045 | **0.358±0.046** | 0.397±0.042 | 0.457±0.044 | 0.503±0.045 | **0.524±0.047** | 0.681±0.052 |
| | MOIRAI | 0.442±0.038 | **0.350±0.047** | **0.356±0.044** | 0.433±0.043 | 0.473±0.039 | 0.524±0.042 | 0.568±0.041 | 0.583±0.040 | 0.649±0.053 |
| | MOMENT | 0.776±0.055 | 0.650±0.045 | 0.589±0.050 | 0.582±0.046 | 0.665±0.045 | 0.987±0.063 | 0.914±0.076 | 0.826±0.073 | 0.706±0.058 |
| Climate (TC-PRIMED) | Mean | 0.688±0.015 | 0.688±0.015 | 0.688±0.015 | 0.688±0.015 | 0.688±0.015 | 0.688±0.016 | 0.688±0.016 | – | – |
| | LOCF | **0.430±0.010** | **0.424±0.009** | **0.434±0.009** | **0.488±0.011** | **0.489±0.011** | **0.560±0.014** | **0.569±0.014** | – | – |
| | Chronos | 0.436±0.010 | 0.464±0.010 | 0.442±0.009 | 0.489±0.012 | 0.518±0.012 | 0.675±0.016 | 0.688±0.015 | – | – |
| | MOIRAI | 0.447±0.009 | 0.448±0.009 | 0.466±0.011 | **0.488±0.012** | 0.495±0.012 | 0.676±0.015 | 0.688±0.015 | – | – |
| | MOMENT | 0.513±0.011 | 0.735±0.012 | 0.530±0.012 | 0.539±0.014 | 0.530±0.013 | 0.676±0.015 | 0.688±0.016 | – | – |

*Table 3.* Relative forecasting NMAE $R(g) = \frac{1}{|\mathcal{K}|} \sum_{k \in \mathcal{K}} \frac{1}{|S_k|} \sum_{s \in S_k} \frac{\mathcal{L}_{k,s}^{\text{NMAE}}(g)}{\mathcal{L}_{k,s}^{\text{NMAE}}(\text{native})}$ per (domain, model, signal speed). Coarsening mapping as in Table 1. Native condition ($R = 1.00$ by construction) and Mean baseline ($R_{\text{mean}} = 1.00$ by construction) are omitted. Cells show $\mu \pm h$ where $h$ is the 95 % percentile-bootstrap half-width over test stays ($B = 1,000$).

| Domain | Model | Speed | $g_1$ | $g_2$ | $g_3$ | $g_4$ | $g_5$ | $g_6$ | $g_7$ | $g_8$ |
|---|---|---|---|---|---|---|---|---|---|---|
| Healthcare (MIMIC-IV) | LOCF | fast | 1.04±0.02 | 1.12±0.01 | 1.35±0.02 | 1.59±0.02 | 1.37±0.02 | 2.39±0.05 | 2.39±0.05 | – |
| | LOCF | slow | **1.02±0.01** | **1.05±0.01** | **1.25±0.03** | 1.59±0.05 | 1.61±0.05 | 2.90±0.18 | 2.90±0.17 | – |
| | Chronos | fast | 1.14±0.01 | 1.06±0.01 | **1.32±0.01** | 1.64±0.02 | 1.41±0.01 | 2.08±0.03 | 2.08±0.03 | – |
| | Chronos | slow | 1.47±0.03 | 1.15±0.02 | 1.26±0.03 | 1.83±0.06 | 1.56±0.05 | 2.03±0.08 | 2.03±0.08 | – |
| | MOIRAI | fast | 1.09±0.01 | 1.15±0.01 | 1.62±0.01 | 1.60±0.02 | 1.68±0.02 | 1.86±0.02 | 1.86±0.02 | – |
| | MOIRAI | slow | 1.28±0.05 | 1.33±0.06 | 1.92±0.10 | 1.75±0.08 | 2.33±0.13 | **1.71±0.08** | **1.71±0.08** | – |
| | MOMENT | fast | **1.00±0.01** | 0.96±0.01 | 1.32±0.01 | 1.38±0.01 | 1.07±0.01 | 1.44±0.02 | 1.44±0.02 | – |
| | MOMENT | slow | 1.75±0.09 | 1.46±0.06 | 1.56±0.09 | **1.53±0.07** | **1.36±0.06** | 1.78±0.08 | 1.78±0.08 | – |
| Fusion (MAST) | LOCF | fast | **1.00±0.01** | **1.00±0.01** | 1.03±0.01 | 1.06±0.01 | 1.17±0.01 | 1.39±0.05 | 1.41±0.05 | **3.04±0.21** |
| | LOCF | slow | 1.07±0.11 | 1.07±0.11 | 1.09±0.10 | 1.12±0.10 | 1.25±0.10 | 1.50±0.14 | 2.27±0.67 | 3.46±0.91 |
| | Chronos | fast | 1.19±0.03 | 1.02±0.01 | 1.16±0.04 | 1.15±0.05 | 1.65±0.12 | 2.11±0.21 | 1.97±0.12 | 5.61±0.68 |
| | Chronos | slow | 0.91±0.03 | 1.01±0.06 | 0.94±0.04 | 1.06±0.04 | 1.17±0.08 | 1.39±0.12 | 1.64±0.16 | 1.71±0.17 |
| | MOIRAI | fast | 1.16±0.04 | 1.01±0.02 | 1.62±0.11 | 1.59±0.09 | 1.91±0.10 | 2.21±0.13 | 2.52±0.15 | 5.08±0.44 |
| | MOIRAI | slow | 0.90±0.05 | 0.95±0.04 | 1.05±0.04 | 1.11±0.07 | 1.15±0.07 | 1.38±0.13 | 1.55±0.21 | 1.72±0.18 |
| | MOMENT | fast | 1.17±0.05 | 1.09±0.03 | 1.39±0.03 | 1.67±0.05 | 2.84±0.10 | 2.05±0.07 | 2.11±0.06 | 3.78±0.19 |
| | MOMENT | slow | **0.89±0.05** | **0.79±0.05** | **0.77±0.05** | **0.85±0.04** | **1.07±0.06** | **1.17±0.10** | **1.11±0.11** | **0.98±0.09** |
| Climate (TC-PRIMED) | LOCF | slow | **1.02±0.01** | 1.12±0.02 | 1.38±0.03 | 1.39±0.03 | **1.78±0.04** | **1.82±0.04** | – | – |
| | Chronos | slow | 1.17±0.01 | 1.13±0.01 | 1.32±0.02 | 1.42±0.02 | 2.96±0.08 | 3.04±0.08 | – | – |
| | MOIRAI | slow | 1.07±0.01 | 1.10±0.01 | 1.20±0.02 | 1.22±0.02 | 2.58±0.07 | 2.65±0.07 | – | – |
| | MOMENT | slow | 1.79±0.02 | **1.08±0.01** | **1.10±0.01** | **1.08±0.01** | 2.21±0.06 | 2.26±0.06 | – | – |

*Table 4.* Relative forecasting NMAE $R(g) = \frac{1}{|\mathcal{K}|} \sum_{k \in \mathcal{K}} \frac{1}{|S_k|} \sum_{s \in S_k} \frac{\mathcal{L}_{k,s}^{\text{NMAE}}(g)}{\mathcal{L}_{k,s}^{\text{NMAE}}(\text{native})}$ aggregated over fast and slow signal speeds. Mapping and CI as in Table 3.

| Domain | Model | $g_1$ | $g_2$ | $g_3$ | $g_4$ | $g_5$ | $g_6$ | $g_7$ | $g_8$ |
|---|---|---|---|---|---|---|---|---|---|
| Healthcare (MIMIC-IV) | LOCF | **1.03±0.02** | **1.09±0.01** | 1.30±0.02 | 1.59±0.03 | 1.49±0.03 | 2.64±0.11 | 2.64±0.11 | – |
| | Chronos | 1.30±0.02 | 1.10±0.01 | **1.29±0.02** | 1.73±0.04 | 1.49±0.03 | 2.06±0.05 | 2.06±0.05 | – |
| | MOIRAI | 1.18±0.03 | 1.24±0.03 | 1.77±0.05 | 1.68±0.05 | 2.00±0.07 | 1.78±0.05 | 1.78±0.05 | – |
| | MOMENT | 1.37±0.05 | 1.21±0.03 | 1.44±0.05 | **1.46±0.04** | **1.22±0.03** | **1.61±0.05** | **1.61±0.05** | – |
| Fusion (MAST) | LOCF | 1.04±0.06 | 1.04±0.06 | 1.06±0.06 | **1.09±0.05** | 1.21±0.06 | 1.45±0.10 | 1.84±0.36 | 3.25±0.56 |
| | Chronos | 1.05±0.03 | 1.02±0.03 | **1.05±0.04** | 1.11±0.05 | 1.41±0.10 | 1.75±0.17 | 1.81±0.14 | 3.66±0.42 |
| | MOIRAI | **1.03±0.04** | 0.98±0.03 | 1.33±0.07 | 1.35±0.08 | 1.53±0.09 | 1.79±0.13 | 2.04±0.18 | 3.40±0.31 |
| | MOMENT | **1.03±0.05** | **0.94±0.04** | 1.08±0.04 | 1.26±0.05 | 1.96±0.08 | 1.61±0.08 | **1.61±0.08** | 2.38±0.14 |
| Climate (TC-PRIMED) | LOCF | **1.02±0.01** | 1.12±0.02 | 1.38±0.03 | 1.39±0.03 | **1.78±0.04** | **1.82±0.04** | – | – |
| | Chronos | 1.17±0.01 | 1.13±0.01 | 1.32±0.02 | 1.42±0.02 | 2.96±0.08 | 3.04±0.08 | – | – |
| | MOIRAI | 1.07±0.01 | 1.10±0.01 | 1.20±0.02 | 1.22±0.02 | 2.58±0.07 | 2.65±0.07 | – | – |
| | MOMENT | 1.79±0.02 | **1.08±0.01** | **1.10±0.01** | **1.08±0.01** | 2.21±0.06 | 2.26±0.06 | – | – |

*Table 5.* Offline imputation NMAE per (domain, model, signal speed) across grid coarsening levels $g_1$–$g_8$. Coarsening mapping as in Table 1. Foundation models (Chronos, MOIRAI, MOMENT) are causal forecasters and are not evaluated on imputation tasks; only LOCF and Mean baselines apply. Cells show $\mu \pm h$ where $h$ is the 95 % percentile-bootstrap half-width over test stays ($B = 1,000$).

| Domain | Model | native | $g_1$ | $g_2$ | $g_3$ | $g_4$ | $g_5$ | $g_6$ | $g_7$ | $g_8$ |
|---|---|---|---|---|---|---|---|---|---|---|---|
| Healthcare (MIMIC-IV) | Mean | 0.786±0.008 | 0.786±0.008 | 0.786±0.009 | 0.786±0.009 | 0.786±0.009 | 0.786±0.009 | **0.786±0.008** | **0.786±0.008** | – |
| | LOCF | **0.646±0.009** | **0.555±0.008** | **0.557±0.007** | **0.575±0.008** | **0.631±0.009** | **0.748±0.010** | 0.906±0.014 | 0.906±0.015 | – |
| Fusion (MAST) | Mean | **0.468±0.035** | 0.468±0.032 | 0.468±0.033 | 0.468±0.031 | 0.468±0.033 | 0.468±0.035 | 0.468±0.033 | 0.468±0.030 | **0.468±0.031** |
| | LOCF | 0.736±0.054 | **0.322±0.034** | **0.323±0.034** | **0.325±0.036** | **0.326±0.032** | **0.335±0.035** | **0.344±0.034** | **0.362±0.035** | 0.632±0.062 |
| Climate (TC-PRIMED) | Mean | 0.630±0.018 | 0.630±0.016 | 0.630±0.018 | 0.630±0.017 | 0.630±0.019 | 0.630±0.018 | 0.630±0.017 | – | – |
| | LOCF | **0.396±0.015** | **0.270±0.010** | **0.271±0.010** | **0.275±0.010** | **0.286±0.010** | **0.320±0.012** | **0.418±0.014** | – | – |

*Table 6.* Causal imputation NMAE per (domain, model, signal speed) across grid coarsening levels $g_1-g_8$. Coarsening mapping as in Table 1. Foundation models are not evaluated on imputation tasks; only LOCF and Mean baselines apply. Cells show $\mu \pm h$ where $h$ is the 95 % percentile-bootstrap half-width over test stays ($B = 1{,}000$).

| Domain | Model | native | $g_1$ | $g_2$ | $g_3$ | $g_4$ | $g_5$ | $g_6$ | $g_7$ | $g_8$ |
|---|---|---|---|---|---|---|---|---|---|---|
| Healthcare (MIMIC-IV) | Mean | 0.786±0.008 | 0.786±0.008 | 0.786±0.008 | 0.786±0.008 | **0.786±0.009** | **0.786±0.009** | **0.786±0.009** | **0.786±0.008** | – |
| | LOCF | **0.646±0.010** | **0.618±0.009** | **0.630±0.009** | **0.676±0.010** | 0.906±0.015 | 0.906±0.014 | 0.906±0.015 | 0.906±0.014 | – |
| Fusion (MAST) | Mean | 0.482±0.037 | 0.482±0.037 | 0.482±0.034 | 0.482±0.037 | 0.482±0.037 | 0.482±0.033 | 0.482±0.037 | 0.482±0.034 | **0.482±0.035** |
| | LOCF | **0.383±0.035** | **0.383±0.037** | **0.383±0.035** | **0.385±0.036** | **0.388±0.036** | **0.393±0.037** | **0.410±0.042** | **0.445±0.039** | 0.620±0.096 |
| Climate (TC-PRIMED) | Mean | 0.630±0.018 | 0.630±0.019 | 0.630±0.018 | 0.630±0.018 | 0.630±0.019 | 0.630±0.017 | 0.630±0.017 | – | – |
| | LOCF | **0.396±0.015** | **0.401±0.017** | **0.410±0.015** | **0.419±0.016** | **0.436±0.017** | **0.466±0.018** | **0.565±0.023** | – | – |

*Table 7.* Cross-modal NMAE per (domain, model, signal speed) across grid coarsening levels $g_1-g_8$. Coarsening mapping as in Table 1. Cross-modal partitions source and target signal sets to be disjoint, so LOCF cannot use a predecessor of a target signal in context — it falls back to the training-mean penalty, matching the Mean baseline by construction. Cells show $\mu \pm h$ where $h$ is the 95 % percentile-bootstrap half-width over test stays ($B = 1{,}000$).

| Domain | Model | native | $g_1$ | $g_2$ | $g_3$ | $g_4$ | $g_5$ | $g_6$ | $g_7$ | $g_8$ |
|---|---|---|---|---|---|---|---|---|---|---|
| Healthcare (MIMIC-IV) | Mean | 0.807±0.005 | 0.807±0.005 | 0.807±0.005 | 0.807±0.005 | 0.807±0.005 | 0.807±0.005 | 0.807±0.005 | 0.807±0.004 | – |
| | LOCF | 0.807±0.005 | 0.807±0.005 | 0.807±0.005 | 0.807±0.005 | 0.807±0.005 | 0.807±0.005 | 0.807±0.005 | 0.807±0.004 | – |
| Fusion (MAST) | Mean | 0.545±0.043 | 0.545±0.039 | 0.545±0.038 | 0.545±0.043 | 0.545±0.040 | 0.545±0.040 | 0.545±0.044 | 0.545±0.044 | 0.545±0.041 |
| | LOCF | 0.545±0.040 | 0.545±0.041 | 0.545±0.040 | 0.545±0.041 | 0.545±0.043 | 0.545±0.040 | 0.545±0.039 | 0.545±0.042 | 0.545±0.040 |
| Climate (TC-PRIMED) | Mean | 0.719±0.016 | 0.719±0.016 | 0.719±0.017 | 0.719±0.015 | 0.719±0.015 | 0.719±0.016 | 0.719±0.017 | – | – |
| | LOCF | 0.719±0.016 | 0.719±0.017 | 0.719±0.016 | 0.719±0.016 | 0.719±0.016 | 0.719±0.016 | 0.719±0.015 | – | – |

## C. Baseline Scores

We evaluate the mean and LOCF baselines across the full benchmark to establish lower-bound reference scores for future models.

*Table 8.* Baseline NMAE and NRMSE scores at native resolution, broken down by domain and task. Scores use the five-level hierarchical aggregation (signal channel → sample → task → domain → overall). Lower is better; a score of 1.0 corresponds to predicting the training mean at every query point. Domain scores are the mean over the four tasks; the overall score is the mean over the three domains.

| Model | Metric | Fusion | | | | | TC-Primed | | | | | Healthcare | | | | | Overall |
|---|---|---|---|---|---|---|---|---|---|---|---|---|---|---|---|---|---|
| | | Fcast. | Causal | Offline | X-Modal | Dom. | Fcast. | Causal | Offline | X-Modal | Dom. | Fcast. | Causal | Offline | X-Modal | Dom. | |
| LOCF | NMAE | 0.262 | 0.383 | 0.736 | 0.545 | 0.481 | 0.430 | 0.396 | 0.396 | 0.719 | 0.485 | 0.448 | 0.646 | 0.646 | 0.807 | 0.637 | 0.534 |
| | NRMSE | 0.305 | 0.424 | 0.824 | 0.659 | 0.553 | 0.500 | 0.496 | 0.496 | 0.823 | 0.579 | 0.516 | 0.749 | 0.749 | 0.867 | 0.720 | 0.617 |
| Mean | NMAE | 0.474 | 0.483 | 0.468 | 0.545 | 0.493 | 0.688 | 0.630 | 0.630 | 0.719 | 0.667 | 0.753 | 0.786 | 0.786 | 0.807 | 0.783 | 0.647 |
| | NRMSE | 0.505 | 0.517 | 0.545 | 0.659 | 0.556 | 0.732 | 0.675 | 0.675 | 0.823 | 0.726 | 0.807 | 0.877 | 0.877 | 0.867 | 0.857 | 0.713 |

*Table 9.* Benchmark task settings per domain and task. $t_c$ is the forecasting cut-off time; $h = [\tau_{\text{low}}, \tau_{\text{high}}]$ is the forecast horizon band relative to $t_c$; $\mathcal{H}$ is the held-out signal set for imputation tasks; $[t_a, t_b]$ is the masked interval for offline imputation; $t_f^{(s)}$ is the failure time and $t_p$ the present time for causal imputation; $\mathcal{S}_{\text{src}}$ and $\mathcal{S}_{\text{tgt}}$ are the source and target signal sets for cross-modal prediction. All times are given in domain-natural units.

| Task | Parameter | Fusion (MAST) | Healthcare (MIMIC-IV) | Climate (TC-Primed) |
|---|---|---|---|---|
| Forecasting | $t_c$ | 150 ms | 12 h | 24 h |
| | $h = [\tau_{\text{low}}, \tau_{\text{high}}]$ | [0, 50 ms] | [0, 6 h] | [0, 24 h] |
| Offline Imputation | $\mathcal{H}$ | equilibrium reconstruction (18 signals) | blood pressure (1 signal) | satellite overpass scalars (4 signals) |
| | $[t_a^{(s)}, t_b^{(s)}]$ | [150, 350] ms | [6, 12] h | [24, 72] h |
| Causal Imputation | $\mathcal{H}$ | equilibrium reconstruction (18 signals) | blood pressure (1 signal) | satellite overpass scalars (4 signals) |
| | $t_f^{(s)}$ | 250 ms | 6 h | 24 h |
| | $t_p$ | 400 ms | 12 h | 72 h |
| Cross-Modal Prediction | $\mathcal{S}_{\text{src}}$ | magnetics, diagnostics, PF coils (21 signals) | vitals: HR, SpO$_2$, RR, BP, Temp (5 signals) | track & reanalysis env. (12 signals) |
| | $\mathcal{S}_{\text{tgt}}$ | equilibrium reconstruction (18 signals) | neurological: GCS (1 signal) | satellite overpass scalars (4 signals) |

## D. Related Work

Here, we conduct a more thorough evaluation of existing benchmarks and time series foundation models. We focus on both general time series and single-domain scientific time series.

**Time Series Benchmarking Infrastructure**   While rich, the benchmarking infrastructure for time series has converged to a structurally uniform data model in which observations are pre-aligned to a regular temporal grid before any model sees them. Large-scale repositories such as the Monash Time Series Forecasting Repository(Godahewa et al., 2021), collect multiple datasets spanning different frequencies by treating each dataset as a fixed frequency sequence and evaluating models on different forecasting horizons. TSLib (Wu et al., 2023; Wang et al., 2024) expands on this by testing models across more time-series analysis tasks, including imputation and anomaly detection. However, it remains grid-committed and its imputation task masks values within an already-aligned tensor rather than ingesting genuinely irregular timestamps.

Recently, the rise of pre-trained foundation models has prompted the development of benchmarks that are specifically designed for zero-shot and few-shot evaluation. GIFT-Eval (Aksu et al., 2024) has become widely adopted. It curates 23 datasets comprising 144,000 series and 177 million data points across seven domains and ten frequencies, alongside an explicit pre-training corpus of 88 datasets and 230 billion data points to mitigate leakage. Many new benchmarks have since been designed to evaluate time series foundation models, see (Qiao et al., 2026) for an in-depth review of the history of these benchmarks and some further limitations. Yet every one of the aforementioned benchmarks shares the same structural commitment that motivates our work. That is, all observations are aligned to a declared sampling frequency before evaluation begins, and the choice of that frequency is treated as a benchmark design constant rather than a modelling variable. As demonstrated in (Frost et al., 2026; Kazijevs & Samad, 2023a) and in our empirical results, preprocessing choices dictate downstream model rankings.

The closest general time-series benchmark we could find that attempts to move away from the grid assumption is Time-IMM (Chang et al., 2025). Time-IMM preserves genuine irregularity in datasets and incorporates text data as a second modality. While they also target irregular multivariate forecasting, they define 'multimodal' as numerical-plus-text fusion and benchmark fusion strategies. Our benchmark instead targets multimodal as heterogeneous signal types within scientific instrumentation data, isolates grid-commitment as a benchmarking confound, and evaluates four tasks, including causal imputation and cross-modal prediction at native resolution.

**Domain-Specific Benchmarks**   While, as discussed above, existing general time-series benchmarks remain limited to the grid assumption, scientific domains have independently developed bespoke evaluation infrastructure. Tokamark (Rousseau et al., 2026) is a recent work that provides the first general benchmark in the fusion domain. It uses the same FAIR-MAST (Jackson et al., 2024; 2025) dataset and signals that we use and retains the irregular, heterogeneous and multimodal structure that is apparent in fusion data. Tokamark provides the infrastructure for advancing fusion-specific models. However, it is still in its infancy, with only a baseline model and Tokamind (Boschi et al., 2026) evaluated.

In climate science, the benchmarking literature is more saturated than in fusion. WeatherBench2 (Rasp et al., 2023) remains the standard benchmark in climate science. However, much like the general time-series benchmarks, WeatherBench 2 strictly enforces a uniform grid structure. To evaluate different models fairly, the framework explicitly pre-aligns the data by re-gridding all model forecasts and ground truth observations to a standardised, common spatial resolution before computing any metrics. Recent benchmarks like AtmosArena (Nguyen et al., 2024), WxC-Bench (Shinde et al., 2024) and ClimateBench-M (Fu et al., 2025) build on WeatherBench2 by adding downstream tasks and multimodality. Yet they all avoid ingesting irregular, native scientific data. In contrast, the TC PRIMED dataset (Razin et al., 2023) used in our work includes irregular satellite overpasses that are incompatible with a regular grid.

In healthcare, there are few existing benchmarks for time series data. Namely, the HiRID-ICU-Benchmark (Yèche et al., 2022) and the YAIB(Yet Another ICU Benchmark)(van de Water et al., 2024). They both focus on predictive tasks derived from intensive care unit (ICU) records. The HiRID-ICU-Benchmark evaluates models on classification and regression tasks using a single, high-resolution ICU dataset. YAIB expands this scope by providing a modular, multi-centre framework that brings together several open-access ICU datasets (including MIMIC) into a unified format for standardised tasks. However, despite their clinical utility, both benchmarks strictly enforce a fixed temporal grid structure. To process naturally irregular and event-based clinical measurements, the HiRID-ICU-Benchmark employs a "High-Resolution Gridding" step that explicitly resamples all observations into a uniform 5-minute resolution grid, populating intervals with the last measured value. Similarly, YAIB's default preprocessing pipeline forces asynchronous data onto a regular temporal grid by binning

observations into fixed intervals and applying forward-filling imputation to handle missingness. Consequently, models evaluated on these benchmarks process time series as pre-aligned arrays rather than learning directly from genuine, unaligned clinical timestamps. Other benchmarks, such as CLIMB (Dai et al., 2025), build on earlier work, such as BenchMD (Wantlin et al., 2023), by incorporating data from 13 clinical domains spanning multiple modalities. These modalities include time series data; however, they place greater emphasis on medical imaging than on ICU time series data.

## E. Datasets

Here we provide tables describing the datasets used in the benchmark. We detail the number of samples in each split, the units and modalities of each signal, as well as the typical sampling rates and the percentile range of sample durations across the dataset (p10–p90). For healthcare, the data is split randomly into the test, train and validation sets by ICU stay. In fusion and climate, they are split chronologically by shot/storm, as models trained on fusion/weather data are typically trained on historical data and, in fusion, plasma discharges exhibit non-stationary distribution drift driven by experimental evolution and hardware wear-and-tear (Rajput et al., 2025).

*Table 10.* **Healthcare dataset (MIMIC-IV).** Each sample is one ICU stay ($\geq$24 h), split deterministically by ICU stay. Train / val / test: 52 380 / 11 224 / 11 225 (total: 74 829 stays). Median duration 67 h (p10–p90: 26 h–323 h). Signal modalities: 4 scalar, 2 multi-channel scalar. All signals are asynchronous point-in-time scalars with irregular timestamps.

| Group | Signal | Category | Unit | Channels | Typical rate |
|---|---|---|---|---|---|
| Vital signs | Heart rate | Scalar | bpm | 1 | $\sim$1 h |
| | SpO$_2$ | Scalar | % | 1 | $\sim$1 h |
| | Respiratory rate | Scalar | breaths min$^{-1}$ | 1 | $\sim$1 h |
| | Blood pressure | Scalar (multi-ch) | mmHg | 3 | $\sim$1 h |
| | Temperature | Scalar | °C | 1 | $\sim$4 h |
| Neurological | GCS | Scalar (multi-ch) | score | 3 | $\sim$4 h |
| **Total** | | | | **10** | |

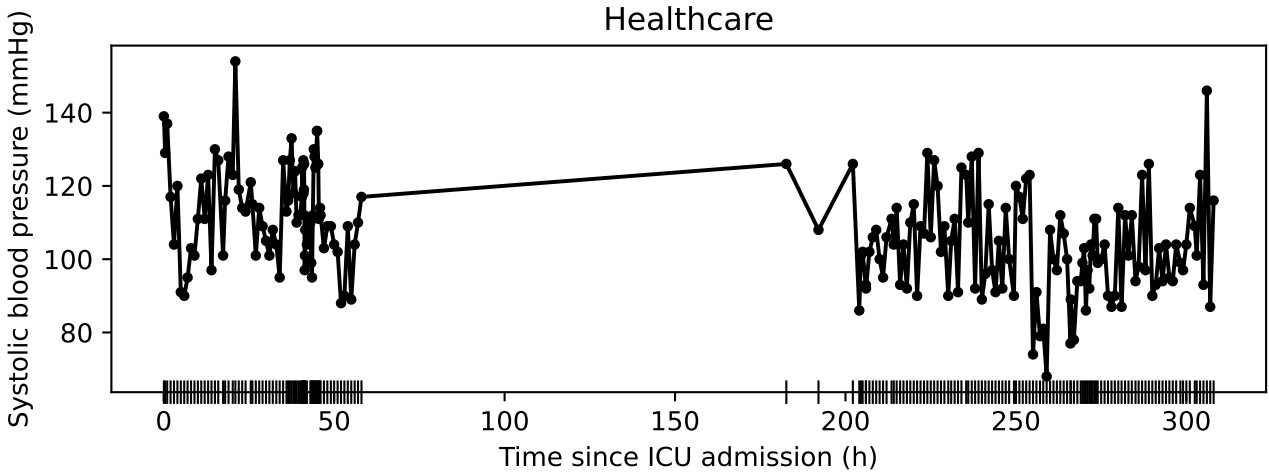

*Figure 3.* Visualisation of the irregular sampling present in the MIMIC III dataset. Lines on the x-axis display the sample density in the Systolic blood pressure.

*Table 11.* **Fusion dataset (MAST tokamak).** We use a subset of shots with the same signals as in the Tokamind ([Boschi et al.](), 2026) benchmark. Each sample is one plasma discharge (shot). Train / val / test: 2 022 / 433 / 434 (total: 2 889 shots). Median duration 538 ms (p10–p90: 417 ms–773 ms). Signal modalities: 20 scalar, 14 multi-channel scalars, 4 profile-1D, 1 2D field. Signals span five orders of magnitude in sampling rate (200 Hz to 500 kHz), making this a strongly multi-rate dataset.

| Subsystem | Signal | Category | Unit | Channels | Sampling rate |
|---|---|---|---|---:|---|
| Magnetics | Flux loop flux | Scalar (multi-ch) | Wb | 15 | 5 kHz |
| | CCBV poloidal B-field | Scalar (multi-ch) | T | 40 | 5 kHz |
| | OBR poloidal B-field | Scalar (multi-ch) | T | 19 | 5 kHz |
| | OBV poloidal B-field | Scalar (multi-ch) | T | 19 | — |
| | Saddle probe voltage | Scalar (multi-ch) | V | 12 | 50 kHz |
| | CC toroidal B-field | Scalar (multi-ch) | T | 3 | 500 kHz |
| | OMV poloidal voltage | Scalar (multi-ch) | V | 3 | 500 kHz |
| Plasma diagnostics | Thomson $T_e$ (profile) | Profile 1D | eV | 1 | 200 Hz |
| | Thomson $n_e$ (profile) | Profile 1D | $m^{-3}$ | 1 | 200 Hz |
| | Interferometer $n_e$ (line) | Scalar | $m^{-2}$ | 1 | 4 kHz |
| | D$\alpha$ spectrometer | Scalar (multi-ch) | V | 3 | 50 kHz |
| | Soft X-ray (lower camera) | Scalar (multi-ch) | V | 18 | 50 kHz |
| | Soft X-ray (upper camera) | Scalar (multi-ch) | V | 18 | 50 kHz |
| PF coil system | PF coil current | Scalar (multi-ch) | A | 10 | 4 kHz |
| | Solenoid current | Scalar | A | 1 | 4 kHz |
| | PF coil voltage | Scalar (multi-ch) | V | 4 | 4 kHz |
| | Scheduled $I_p$ | Scalar | A | 1 | 4 kHz |
| | Scheduled $n_e$ (line) | Scalar | $m^{-2}$ | 1 | 4 kHz |
| Summary scalars | Plasma current $I_p$ | Scalar | A | 1 | 4 kHz |
| | NBI heating power | Scalar | W | 1 | 4 kHz |
| | Gas injection | Scalar | count | 1 | 4 kHz |
| Equilibrium | Elongation $\kappa$ | Scalar | — | 1 | 200 Hz |
| | Elongation (axis) $\kappa_{ax}$ | Scalar | — | 1 | 200 Hz |
| | Upper triangularity $\delta_u$ | Scalar | — | 1 | 200 Hz |
| | Lower triangularity $\delta_l$ | Scalar | — | 1 | 200 Hz |
| | X-point $R$ | Scalar (multi-ch) | — | 2 | 200 Hz |
| | X-point $Z$ | Scalar (multi-ch) | — | 2 | 200 Hz |
| | Minor radius $a$ | Scalar | m | 1 | 200 Hz |
| | Magnetic axis $R$ | Scalar | m | 1 | 200 Hz |
| | Magnetic axis $Z$ | Scalar | m | 1 | 200 Hz |
| | Safety factor $q_{95}$ | Scalar | — | 1 | 200 Hz |
| | Toroidal $\beta$ | Scalar | — | 1 | 200 Hz |
| | Poloidal $\beta$ | Scalar | — | 1 | 200 Hz |
| | Normalised $\beta_N$ | Scalar | T | 1 | 200 Hz |
| | $B_{vac}$ at $R_{mag}$ | Scalar | T | 1 | 200 Hz |
| | $B_\phi$ at $R_{mag}$ | Scalar | T | 1 | 200 Hz |
| | LCFS contour $R$ (profile) | Profile 1D | m | 1 | 200 Hz |
| | LCFS contour $Z$ (profile) | Profile 1D | m | 1 | 200 Hz |
| | Poloidal flux $\psi$ (2D field) | 2D field | $Wb\,rad^{-1}$ | 1 | 200 Hz |
| **Total** | | | | **193** | |

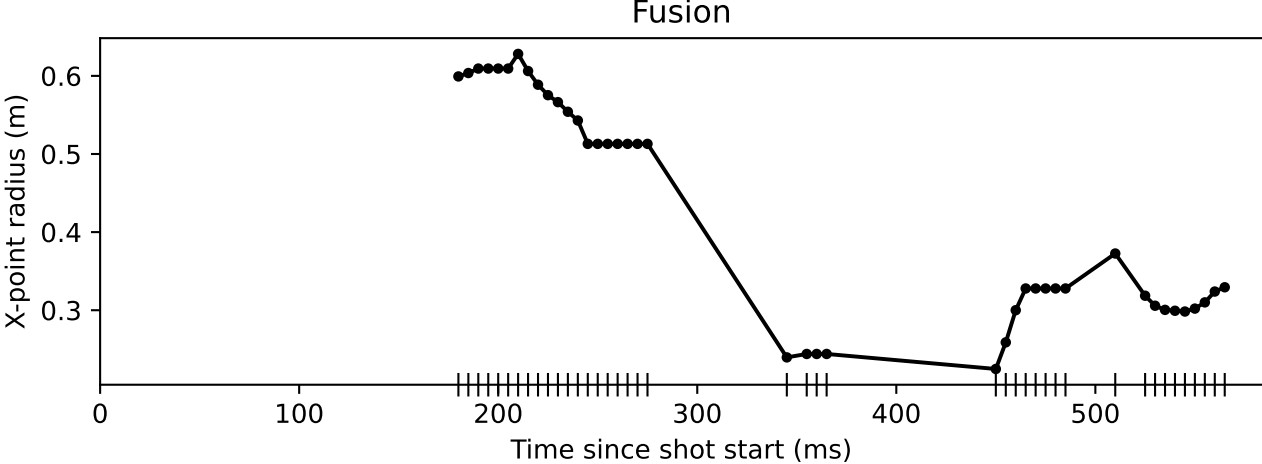

*Figure 4.* Visualisation of the irregular sampling present in the MAST dataset. The values on the x-axis demonstrate the irregular spacing between samples.

*Table 12.* **Tropical cyclone dataset (TC-PRIMED).** Each sample is one tropical cyclone track. Train / val / test: 2 485 / 532 / 534 (total: 3 551 storms). Median duration 162 h (p10–p90: 76 h–307 h). Signal modalities (excluding static context): 11 scalar, 5 multi-ch scalar, 4 2D field. Two temporal sources: regular environmental track data (6 h) and irregularly-timed satellite overpasses (∼45 min median).

| Source | Signal | Category | Unit | Channels | Rate |
|---|---|---|---|---|---|
| Track | Intensity | Scalar | kt | 1 | 6 h (regular) |
| | Min. pressure | Scalar | hPa | 1 | 6 h (regular) |
| | Storm speed | Scalar | $m\,s^{-1}$ | 1 | 6 h (regular) |
| | Storm heading | Scalar | deg | 1 | 6 h (regular) |
| Env. diagnostics | Sea surface temp. | Scalar | K | 1 | 6 h (regular) |
| | Max. potential intensity | Scalar | kt | 1 | 6 h (regular) |
| | Wind shear magnitude | Scalar (multi-ch) | $m\,s^{-1}$ | 8 | 6 h (regular) |
| | Wind shear direction | Scalar (multi-ch) | deg | 8 | 6 h (regular) |
| | Precipitable water | Scalar (multi-ch) | $kg\,m^{-2}$ | 5 | 6 h (regular) |
| | Warm core anomaly | Scalar (multi-ch) | K | 23 | 6 h (regular) |
| | Cyclone phase ($B$) | Scalar | m | 1 | 6 h (regular) |
| | Vertical velocity | Scalar (multi-ch) | $m\,s^{-1}$ | 4 | 6 h (regular) |
| Overpass scalars | Intensity | Scalar | kt | 1 | ∼45 min (irregular) |
| | Min. pressure | Scalar | hPa | 1 | ∼45 min (irregular) |
| | Storm speed | Scalar | $m\,s^{-1}$ | 1 | ∼45 min (irregular) |
| | Distance to land | Scalar | km | 1 | ∼45 min (irregular) |
| Overpass imagery | Surface precip. (GPROF) | 2D field | $mm\,hr^{-1}$ | 1 | ∼45 min (irregular) |
| | Ice water path (GPROF) | 2D field | $kg\,m^{-2}$ | 1 | ∼45 min (irregular) |
| | IR brightness temp. | 2D field | K | 1 | ∼45 min (irregular) |
| | Radar precip. rate | 2D field | $mm\,hr^{-1}$ | 1 | ∼45 min (irregular) |
| **Total** | | | | **63** | |

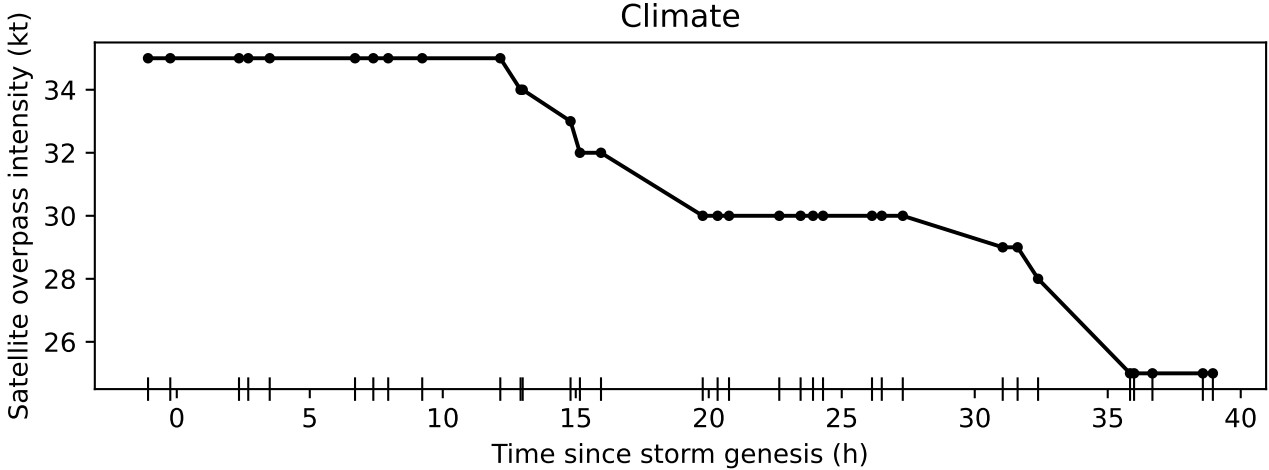

*Figure 5.* Visualisation of the irregular sampling present in the TCPrimed dataset. The values on the x-axis demonstrate the irregular spacing between samples.

