# OpenReview forum: "Towards Benchmarking Time Series Foundation Models on Native Scientific Data"
_ICML.cc/2026/Workshop/FMSD — FMSD @ ICML 2026 Poster_

### Official Review · Reviewer_Qee3 · 2026-05-19
**good writing and topic but questionable method**

**Rating:** 5
**Confidence:** 4

**Review:**

## Summary

The paper proposes a new benchmark for irregular sampled time series, especially scientific data. The need for such a benchmark is motivated by underrepresentation in existing benchmarks and by the influence of pre-processing (transformation to evenly distributed timestamps) to benchmark results.
The proposed benchmark consists of 3 datasets covering nuclear fusion, healthcare, and climate, together with 4 different prediction tasks, whose results are evaluated in a hierarchical manner. In a first step three Time Series Foundation Models are benchmarked on the forecasting task on different grid resolutions showing degradation on lower resolutions. In this forecasting task, the test set is kept at irregular original timestamps while the context is resampled and aggregated from original high resolution time stamps to very low resolution timestamps. The paper concludes that performance degrades by this grid alignment, which is invisible in conventional benchmarks.

---

## Strengths

- the paper is well written with detailed explanations of the approaches
- it is a very timely and interesting area: irregular timestamps are not only in scientific data present, but also in a lot of other domains. This topic is underrepresented in the current benchmarks.
- all steps are described and most upcoming questions during reading answered themself
- the choice of metrics is a very challenging task, and the authors recognize the problems of comparability by introducing two scaled metrics and are calculating on different hierarchical levels
- the tasks itself deliver a more holistic evaluation than most existing benchmarks

---

## Areas for Improvement

-  the metrics still have problems when averages are near zero. There is no easy solution for this, but including a baseline-normalized metric like MASE should be considered for a benchmark, where one can directly see the performance relative to a naive baseline and where you wouldn't have the division by zero problem mentioned on page 3.
- in general: every aggregation step (e.g. going from irregular timestamps to 15 min data to 1 hour data) is a step of information loss. When comparing, each approach should have the same information for their prediction task. In this case the mean baseline has other information than the other approaches. The baseline is in this paper always the average across the irregular timestamps but this average is different than an average across the resampled timestamps. In my opinion it is not a fair comparison when a model only has less informative data (the aggregated context) and compare it against a baseline which is based on the most informative data (even if it is a very dumb baseline).
- the evaluation is described like following and the results are also implying the same: the chosen evaluation setting is aggregating context while keeping the test set at original resolution/timestamps. This is highly artificial and the results are not surprising. It is practically not possible that the models could perform better as less information is provided in the context (see above: aggregating is a form of information loss), otherwise it would be simply luck. *For each test sample, the forecasting query set and evaluation metric are held fixed while only the context representation is varied, from native irregular timestamps to increasingly coarse uniform grids.* From where should the model infer what happens at lower granularity when never seen it? To put it simply: in its extreme form the evaluation setting is about providing daily data (one timestamp per day) and asking for a forecast for the next 30 seconds. Also without seeing the code implementation I am unsure if the TSFMs are even capable of providing the asked prediction points as they are usually working with the same regular timestamps across context and forecast horizon (15 min data in, 15 min data out).
- an investigation is missing if the proposed test datasets are included in the pre-training data of the Time Series Foundation Models
- Time Series Foundation Models are tested only univariate although modern variants of each (chronos-2, TimesFM2.5,...) can handle covariates
- in the causal imputation task is not clear what should be really "causal", I understand it simply as a forecasting task (with forecast start at tp-(Hmin)) with future known covariates

---

## Detailed Comments

If there is a misunderstanding in the evaluation set up of the TSMFs (which I don't think because it is basically described so), I encourage to write it differently, so the reader understands it easier. Because what you are actually showing is, that the models are not capable of understanding the underlying ground truth process on a coarser grid. But this is not a problem of the grid itself. Guessing the underlying ground truth on sparse data requires some external knowledge because the provided data itself doesn't hold the information for a higher resolution forecast than provided context. In theory the foundation models could have this external knowledge because of their pre-training (obviously they don't have it yet), but this is a totally different story and not an alignment problem of existing benchmarks.

---

## Justification of Score

Irregular timestamps are a very timely topic and not seen enough in the community, and there is definitely a lack of suitable benchmarks. The general setup, writing and literature work is appealing, but I see a logical mismatch in the hypothesis and how to prove it.

---

### Official Review · Reviewer_SmwB · 2026-05-20

**Rating:** 8
**Confidence:** 3

**Review:**

## Summary
The authors propose a cross-domain benchmark designed to evaluate time series foundation models on native, unaligned scientific data. The paper addresses a structural commitment inherent in existing benchmarks: pre-aligning observations to a uniform grid before evaluation, which discards irregular, multi-rate structures common in domains like healthcare, climate, and nuclear fusion. By introducing a unified observation set format and a hierarchical metric system, the authors conduct grid-sweep experiments demonstrating that standard preprocessing induces significant, architecture-specific performance degradation that remains invisible under conventional evaluation protocols. The benchmark defines four tasks: forecasting, offline imputation, causal imputation, and cross-modal prediction.

## Strengths
- Highly Relevant and Underexplored Perspective: The paper tackles a critical but underexplored issue of the hidden costs of temporal resampling and grid alignment in scientific time series benchmarking. Treating resampling as a modeling variable rather than a preprocessing constant is a valuable paradigm shift.
- Rigorous and Interesting Empirical Analysis: The diagnostic grid-sweep experiment effectively provides clear evidence that forcing data onto a regular grid degrades performance across different models and scientific domains.
- Well-Structured Benchmark: The benchmark is thoughtfully constructed across diverse domains (nuclear fusion, healthcare, and climate). The proposed unified observation set format effectively preserves native timestamps and missingness, and the hierarchical, scale-normalised metric system successfully prevents high-frequency or data-rich domains from dominating the aggregate scores.

## Areas for Improvement
- Limited Application of Foundation Models: The authors evaluate pre-trained foundation models (Chronos, MOIRAI, MOMENT) solely on the forecasting task, noting that they are causal forecasters unable to condition on future observations. However, these models could be adapted for offline and causal imputation tasks by taking only the available past portion of the channel to reconstruct the missing parts (for MOIRAI could be also only masked the missing part). Restricting them entirely from the imputation and cross-modal tasks limits the comprehensiveness of the benchmark.
- Usage of Outdated Models: The empirical evaluation relies on slightly outdated versions of foundation models. For instance, the paper utilizes Chronos-Bolt instead of the newer Chronos-2, and MOIRAI version 1.0 instead of newer iterations like MOIRAI 2.0. Updating the baselines to reflect the most current state-of-the-art would strengthen the relevance of the findings.
- Inconsistent Table Formatting: The authors should carefully check the tables to ensure they match the stated formatting rules. For instance, the caption for Table 1 in Appendix B explicitly claims that "bold results are the best-performing models for each grid resolution". However, this bolding appears to be missing in the data; as an example, Chronos should be bolded as the best performer for the fast signal speed under the g1 column in the fusion domain.

## Detailed Comments
Consideration for Future Work: As an additional consideration for the model evaluation landscape, you might want to mention or explore recent continuous-time architectures like FlowState. FlowState employs a state space model (SSM) encoder and a functional basis decoder, enabling it to inherently adapt its internal dynamics to varying input scales and sampling rates. Because FlowState can dynamically adjust to irregular time series without requiring exposure to all possible frequencies during training, it aligns perfectly with the native-resolution challenges championed in this benchmark. Including such a model could highlight the potential of architectures designed specifically for the unaligned data formats you propose.

## Justification of Score
This submission strongly aligns with the workshop's focus on foundation models for structured data. The core claim (that grid alignment introduces hidden performance costs) is supported by accurate, convincing, and clear empirical evidence. While there are minor weaknesses regarding the specific foundation models chosen and the scope of their application across the benchmark tasks, the overall contribution is significant, conceptually novel, and will undoubtedly be of interest to the workshop audience.

---

### Official Review · Reviewer_NR8w · 2026-05-21
**The paper introduces non-evenly spaced time series datasets and discusses their implications for forecasting in the context of foundation models. The main results suggest that foundation models trained on evenly spaced datasets degrade their performance on these new benchmarks.**

**Rating:** 6
**Confidence:** 4

**Review:**

## Strengths

- The paper proposes new datasets that are underexplored in the literature, which come from scientific processes.
- The introduction of mathematical notation describing these datasets helps to distinguish them better from the traditional evenly spaced datasets.
- Experiments are well designed and show the limitations of TSFM in these settings.
- Experiments include non-forecasting related tasks, such as imputation, that help to understand the limitations of TSFM better.

## Areas for Improvement

- Even though the paper claims to work with non-evenly spaced data, there are no plots confirming it or exemplifying it.
- The notation adds some clarity, but at the same time, any logical or mathematical result is derived from it. It seems to include it just for narrative purposes.
- To better assess the limitations of TSFM, more baseline models should be included, such as SeasonalNaive.

## Detailed comments

- Include more baselines.
- Show the series you're working with.